# *Helicobacter pylori* Secondary Antibiotic Resistance after One or More Eradication Failure: A Genotypic Stool Analysis Study

**DOI:** 10.3390/antibiotics13040336

**Published:** 2024-04-07

**Authors:** Giuseppe Losurdo, Martino Mezzapesa, Ilaria Ditonno, Mariapaola Piazzolla, Maria Pricci, Bruna Girardi, Francesca Celiberto, Grazia Galeano, Giuseppe Riezzo, Francesco Russo, Andrea Iannone, Enzo Ierardi, Alfredo Di Leo

**Affiliations:** 1Section of Gastroenterology, Department of Precision and Regenerative Medicine and Ionian Area, University of Bari, 70124 Bari, Italy; 2THD s.p.a., 42015 Correggio, Italy; 3Ph.D. Course in Organs and Tissues Transplantation and Cellular Therapies, Department of Precision and Regenerative Medicine and Ionian Area, University of Bari, 70124 Bari, Italy; 4Functional Gastrointestinal Disorders Research Group, National Institute of Gastroenterology IRCCS “Saverio de Bellis”, 70013 Castellana Grotte, Italy

**Keywords:** *Helicobacter pylori*, rescue regimen, antibiotics, antibiotic resistance, PCR

## Abstract

*Helicobacter pylori* (*H. pylori*) antibiotic resistance is the leading cause for unsuccessful eradication therapy. After one or more failures, the chance of encountering secondary antibiotic resistance increases. The aim of this study was to characterize genotypic secondary resistance in a cohort of southern Italian *H. pylori* patients with at least one previous failure. Such patients collected stool samples using a dedicated kit (THD fecal test^TM^), and bacterial DNA was extracted and amplified using RT-PCR. Resistance to clarithromycin, amoxicillin, metronidazole, levofloxacin, and tetracycline was assessed using a high-resolution melting curve. We enrolled 50 patients. A total of 72% of patients failed one previous antibiotic course, 16% failed two, 10% failed three, and 2% failed four. The rate of secondary antibiotic resistance was 16% for clarithromycin, 18% for metronidazole, 14% for amoxicillin, 14% for levofloxacin, and 2% for tetracycline. Among the eight clarithromycin-resistant patients, five (62.5%) previously received a clarithromycin-based regimen. The same rate was 33.3% (3/9) for metronidazole. The only tetracycline-resistant patient had received Pylera. In conclusion, our data seem to show that, even though secondary resistance is not very high, resistance to clarithromycin could be very likely related to previous exposure to this antibiotic.

## 1. Introduction

*Helicobacter pylori* (*H. pylori*) is the leading cause of gastritis, peptic ulcers, and other extra-gastric diseases, including iron deficiency anemia, vitamin B12 deficiency, and idiopathic thrombocytopenic purpura [1,2,3,4]. Furthermore, it is a well-recognized class I carcinogen, involved in the pathogenesis of gastric cancer [5,6,7,8]. The therapy of *H. pylori* is based on the combination of several antibiotics. In first-line treatment, the most commonly used regimens are sequential, concomitant, triple, and bismuth-based quadruple (BQT) therapy [9,10,11]. However, antibiotic resistance is a rising phenomenon, and it may explain most cases of therapy failure. Resistance to clarithromycin, for example, strongly decreases the effectiveness of conventional triple therapy [12,13,14], and this is the reason why current guidelines no longer recommend it in geographical areas with high resistance prevalence, thus suggesting the use of other regimen such as BQT or concomitant [15]. Even resistance to levofloxacin, an antibiotic that is used in second-line treatment, is dramatically increasing [16,17,18], and this may reduce the eradication probability of some rescue regimens.

Indeed, after one failure, second-line regimens are required [19]. Even though empirical rescue therapies are still advocated by guidelines, determination of antibiotic resistance may be a promising strategy to achieve success [20,21]. Indeed, after one failure, onset of novel resistance due to exposure to antibiotics (secondary resistance) is quite frequent and may hamper the eradication of the bacterium [22].

The aim of this study was to characterize genotypic secondary resistances in a cohort of southern Italian *H. pylori* patients with at least one previous failure.

## 2. Results

Fifty patients were enrolled. The male/female ratio was 17/33, and the mean age was 53.3 ± 13.6 years. In total, 72% failed one previous antibiotic course, 16% failed two, 10% failed three, and 2% failed four. The most previously failed regimens were the 3-in-1 pill BQT (Pylera^TM^), in 11 cases, followed by sequential therapy (*n* = 9), triple therapy with clarithromycin (*n* = 7), concomitant therapy (*n* = 5), and triple therapy with levofloxacin (*n* = 3). Some other patients had undergone other therapies not listed by the current guidelines (e.g., triple therapy with only amoxicillin and tetracycline or metronidazole, sequential therapy with levofloxacin), while the remaining ones did not recall previous therapies. Three out of fifty patients (6%) had a previous early therapy termination due to side effects.

The rate of secondary antibiotic resistance was 16% for clarithromycin (*n* = 8), 18% for metronidazole (*n* = 9), 14% for amoxicillin (*n* = 7), 14% for levofloxacin (*n* = 7), and 2% for tetracycline (*n* = 1). Multi-resistance was detected in nine patients (18%) and double resistance to clarithromycin + metronidazole was the most common resistance found (*n* = 5). Results are depicted in Figure 1.

Among patients with previous failure to an amoxicillin-containing regimen (Table 1), amoxicillin resistance was found in 4 out of 20 patients (20%). Among patients with previous failure adhering to a clarithromycin-containing regimen, resistance to such an antibiotic was detected in 6 out of 19 patients (31.6%). Among patients with previous failure in adhering to a metronidazole-containing regimen (*n* = 21), metronidazole resistance was found in three cases (14.3%). Failed levofloxacin-based therapy was administered in three patients, but resistance was found in none of them (0%). Finally, among patients with previous failure adhering to a tetracycline-containing therapy, resistance to this antibiotic was detected in 1 out of 12 patients (8.3%), as reported in Table 1.

On the other side, among the seven amoxicillin-resistant patients, three (42.8%) had received amoxicillin-containing therapy. Among the nine amoxicillin-resistant patients, three (33.3%) had received a metronidazole-based regimen. Among the eight amoxicillin-resistant subjects, five (62.5%) had received a clarithromycin-containing regimen. None of the seven levofloxacin-resistant patients had received a therapy based on such an antibiotic. Finally, the only tetracycline-resistant patient had failed Pylera^TM^.

## 3. Discussion

Treatment of *H. pylori* infection is becoming a complex issue worldwide due to the spread of antimicrobial resistance [23]. Argueta et al. in 2021 found a resistance rate for clarithromycin and levofloxacin very close to 30% in the United States, thus confirming what was previously reported in southern European countries and observed in Italy in the last 4 years [24].

As is known, *H. pylori* infection can be diagnosed using non-invasive tests such as of the *H. pylori* antigen in stool samples, a UBT (Urea Breath Test), and serology, as well as invasive tests such as histology for conventional diagnosis, culture, and a PCR (polymerase chain reaction) both for a diagnosis and for the evaluation of antibiotic sensitivity. Invasive tests require endoscopy [25,26,27].

The first susceptibility test has conventionally been based on culturing and susceptibility testing in *H. pylori* isolates, although it is recommended by current guidelines only after repeated treatment failure [15]. Indeed, it is almost impossible to use this method for first-line treatment selection, as a relatively high rate of false-negative results, often resulting in low sensitivity, has thus far weighed on the use of the test on a large scale. This complexity is mainly due to the need to create and maintain a micro-aerophilic environment, in which the bacterium can grow. Other factors that have so far prevented the widespread spread of *H. pylori* culture are the following problems related to the methodology: the number of gastric biopsies, time-consuming endoscopic procedures, conditions, and interval of transportation of biopsy samples, characteristics of the laboratory, long and unpredictable times required to obtain the result of the investigation [25,28,29]. Of note, culturing does not detect the heteroresistant status of *H. pylori*, e.g., the simultaneous presence of sensitive and resistant strains [26,30,31].

As an alternative to bacterial culture and susceptibility testing, techniques based on real-time polymerase chain reaction (RT-PCR) testing (genotypic resistance detection) have been developed [32]. They are based on the principle of amplifying and detecting the point mutations responsible for antibiotic resistance in *H. pylori* DNA isolated from gastric biopsy samples. These culture-free approaches are accurate in revealing minimal traces of resistant genotypic strains as well as uncovering the heteroresistant state. Furthermore, the ability to evaluate resistant mutant genotypes using PCR not only on fresh specimens, but also on archived paraffin-embedded biopsy specimens, which has been shown to provide an equally reliable substrate for DNA analysis as fresh material, has further emphasized the importance and usefulness of these methods [33,34]. Based on what has been reported, molecular tests unquestionably offer advantages and guarantee feasibility compared to cultures even if they are not used in clinical practice. It is presumable that the need for an invasive endoscopic procedure was the most important limit to their diffusion. Therefore, a next step was represented by the attempt to overcome this drawback through an in-depth and appropriate analysis. A further advancement in this area was represented by the possibility of detecting point mutations that confer resistance to antibiotics in fecal samples of bacterial DNA. From 2003 to 2021, several studies were performed on fecal *H. pylori* DNA using a real-time polymerase chain reaction (RT-PCR) for the diagnosis of infection and/or antibiotic susceptibility, and all showed high rates of sensitivity and specificity when culturing or PCRs on gastric biopsy specimens were used as the gold standard [35].

Therefore, genetic mutations conferring resistance can be detected in stools, representing an excellent substrate for studying sensitivity to *H. pylori* antibiotics [36]. Indeed, there is growing evidence of the potential future availability of non-invasive investigations capable of detecting resistance to *H. pylori* antibiotics, such as clarithromycin and quinolones, which have been commonly used until now. These techniques, if performed before first-line therapy, could allow the identification of a subgroup of strains still sensitive to these drugs and, therefore, of patients who can benefit from old regimens (triple and sequential), whose administration has currently been discouraged by the current guidelines [15]. On the other hand, fecal molecular analysis has the advantage of improving patient compliance, reducing the time/cost ratio of the diagnostic procedure and improving the therapeutic outcome [16]. Finally, the potential risk of a future increase in resistance to quadruple regimens, suggested as first-line treatment by the guidelines, as a consequence of their use on a large scale and incomplete patient adherence could be avoided [37].

In the present study, we evaluated, with the use of molecular analysis of fecal specimens, the onset of secondary antibiotic resistance in subjects affected by *H. pylori* infection and subjected to one or more unsuccessful treatment regimens.

Despite the small sample, we observed that secondary resistance rates were almost low. Indeed, the percentage of secondary antibiotic resistance was 18% for metronidazole, 16% for clarithromycin, 14% for amoxicillin, 14% for levofloxacin, and 2% for tetracycline. Multi-resistance was detected in 18% and double resistance to clarithromycin and metronidazole was the most common resistance. Therefore, the overall secondary resistance rates, after the failure of one or more regimens, appears to be lower than expected. For example, in a Chinese series, the secondary resistances to clarithromycin, metronidazole, and levofloxacin were 96.7%, 90.7%, and 93.1%, respectively [38].

When secondary resistances were considered according to previous failing regimens, an unsuccessful clarithromycin-containing regimen was followed by the highest resistance (31.6%). Among patients with previous failure to an amoxicillin-containing regimen, amoxicillin resistance was found in 20% as well as among patients with previous failure in adhering to a metronidazole-containing regimen, resistance to this antibiotic was found in 14.3%. Patients with previous failure in adhering to tetracycline-containing therapy showed resistance to this antibiotic in 8.3% of cases. These results might suggest that clarithromycin use induces secondary resistance more easily than other antibiotics, even if metronidazole assumption also may be followed by secondary resistance onset in a substantial percentage. Previous exposure to clarithromycin is a known risk factor for secondary resistance, Karczewska found an increase in the resistance rate from 21% to 80% after one failed therapy course [39], and a mathematical model predicted that Clarithromycin resistance may originate from the transmission of resistant bacteria in 98.7% of cases, and derives from spontaneous mutations in the other 1.3% [40]. The rates of secondary resistance to this antibiotic are quite heterogeneous, ranging from 27.2% [41] to 82.9% [42], and this underlines how geographical factors may play a role [43]. In our experience, secondary levofloxacin resistance was 14%, close to the 16% of another study [39]. Interestingly, we found that failed levofloxacin-based therapy was not followed by induced resistances to this antibiotic. This result, even if needing to be confirmed in a large sample, could suggest that levofloxacin resistance is the result of a previous quinolone use for reasons other than *H. pylori* eradication. Indeed, it is well known that quinolone use induces cross-resistance. Finally, the low rate of tetracycline resistance is in agreement with most of the current literature [44]. Anyway, the quite low prevalence of secondary resistance may have many explanations. First, three out of fifty patients had previous early therapy termination due to side effects; therefore, in these patients, real secondary resistance may not have occurred. Another reason could be the molecular biology method. PCR with HRM detects every variation in the genetic sequence, but, differently from sequencing (NGS), it does not precisely define the point mutation; therefore, it is less accurate [45,46]. However, NGS is more expensive and not available in all laboratories. Based on this concern, we preferred HRM, which may be rapid and cost effective for a large-scale analysis. Finally, among the limitations, it should be acknowledged that not all genetic mutations may translate into phenotypic resistance [47].

## 4. Materials and Methods

### 4.1. Patients Selection

We enrolled consecutive dyspeptic patients [48] with at least one failure to a previous antibiotic course against *H. pylori* in the period June 2021–June 2023. Failure had to be demonstrated by persistent positivity to a non-invasive test (stool test or urea breath test) performed at least 4 weeks after stopping antibiotics. Patients were enrolled in two centers (the Gastroenterology Unit, University of Bari, and the National Institute of Gastroenterology “S. de Bellis” Research Hospital).

In detail, we enrolled subjects aged > 18 and who were able to express willingness to participate, presenting with dyspeptic symptoms, such as postprandial fullness, early satiation, epigastric pain, and epigastric burning. We excluded patients with history of gastric or extra-gastric cancer and those who were unable to express informed consent or refused to participate. Additional exclusion criteria were therapy with proton pump inhibitors or histamine receptor antagonists within two weeks from enrollment and use of antibiotics or bismuth salts in the previous four weeks. Furthermore, recent chronic diarrhea was another reason for exclusion because it could restrict the proper collection of fecal samples.

Details about eradication regimens, which were assumed by patients, before this study, were recorded.

The study was conducted in agreement with the indications of the Declaration of Helsinki, and the local Ethics Committee approved the protocol (AOU Consorziale Policlinico di Bari, protocol no. 74413, approved 16 November 2016). All patients signed informed consent.

### 4.2. Evaluation of Antibiotic Resistance

A stool sample was collected from all patients using the THD Fecal Test Device (THD s.p.a., Correggio, Reggio Emilia, Italy), which has shown a sensitivity of 90.2% and a specificity of 98.5% at detecting *H. pylori* genetic sequences [49,50]. This device has a filter blocking the real-time polymerase chain reaction (PCR), inhibiting substances like hemoglobin and its degradation products, polysaccharide complexes, heavy metals, and proteins. Furthermore, it eliminates large molecules such as fibers. The treated solution was finally taken from the reservoir and processed for DNA extraction using a QIAamp DNA Stool Minikit (Qiagen, Hilden, Germany). After this last phase, real-time PCR was performed to assess point mutations linked to *H. pylori* resistance to clarithromycin amoxicillin, metronidazole, levofloxacin, and tetracycline as previously described [16,50,51]. In particular, the following genes implied in antimicrobial resistance were amplified: pbp1 for amoxicillin, rdxA/frxA for metronidazole, 23S rRNA for clarithromycin, gyrA for levofloxacin, and 16S rRNA for tetracycline [52].

Real-time PCR followed by high-resolution melting (HRM) was set to detect mutations. Curves produced using HRM were compared to those derived for wild-type, nonmutated genes from susceptible strains of examined genes to detect resistances [53,54,55].

### 4.3. Statistics

Continuous data were expressed as the mean standard deviation, and categorical variables as proportions/percentages. Graphs were drawn using an Excel version for Windows, Microsoft (Microsoft Italia, Milan, Italy).

## 5. Conclusions

In conclusion, our study shows that, even though secondary resistances were not as high as expected, previous use of clarithromycin strongly increased the risk of novel resistance onset; therefore, its re-use should be discouraged. We did not observe new levofloxacin resistance after its use; therefore, it may be hypothesizable that such resistances have been acquired because of previous quinolones’ use, presumably even with the induction of cross resistance. The fact that the resistance rate to tetracycline is quite low may be a comforting finding, presumably due to its poor use in recent years. However, attention towards a careful use of such an antibiotic should be kept, in order to avoid future spread of its resistance [37]. Finally, it should be acknowledged that bismuth salts are an effective weapon against *H. pylori*, as they have an intrinsic antibacterial effect, and they do not elicit any antimicrobial resistance. It has been demonstrated that adding bismuth to an eradication regimen may result in an additional 30–40% of success in resistant infections [56]. Therefore, when available, adding bismuth should always be considered when facing difficult-to-eradicate *H. pylori*.

## Figures and Tables

**Figure 1 antibiotics-13-00336-f001:**
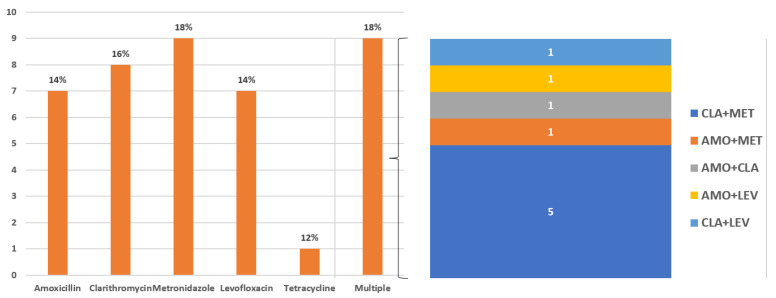
Rates of secondary antibiotic resistance. Multi-resistance is detailed in the right panel of the figure.

**Table 1 antibiotics-13-00336-t001:** Proportions of secondary antibiotic resistance according to the type of previously failed therapy.

Antibiotic	Number of Patients with Secondary Antibiotic Resistance	Number of Patients Who Had Previously Failed a Regimen Containing the Antibiotic Which They Were Resistant to	Rate (%)
**Amoxicillin**	4	20	20%
**Clarithromycin**	6	19	31.6%
**Metronidazole**	3	21	14.3%
**Levofloxacin**	0	3	0%
**Tetracycline**	1	12	8.3%

## Data Availability

Data are contained within the article.

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
