# Peer review of "Helicobacter pylori Secondary Antibiotic Resistance after One or More Eradication Failure: A Genotypic Stool Analysis Study"

_antibiotics, 2024, doi:10.3390/antibiotics13040336_

Round 1

Reviewer 1 Report

Comments and Suggestions for Authors

Interesting  study of a cohort of patients that had failed previous eradication treatment for H. pylori infection. Bacterial resistance represents a major challenge in infection eradication, and it justifies such a study.

The remarkable low prevalence of resistance in this population as noted by the authors requires some additional comments. First, poor patient compliance could explain some of these findings. For instance, if treatment with one or more of the antibiotics use was stopped too early this could paradoxically explain the relative low resistance percentage to a given agent. Second, the resistance was detected, as specified in Methods by real-time PCR followed by high resolution melting (HRM) to detect mutations. Genotypic resistance is not synonymous of phenotypic resistance as it is not always expressed. Perhaps, authors could comment about this possibility in the Discussion.

Nothing is said of bismuth salts activity. H. pylori resistance to these agents is rare and colloidal bismuth subcitrate has been reported to prevent the development of H. pylori resistance to nitronidazole. It would be worth to highlight the importance of bismuth salts in a world of increasing antibiotic resistance.          

Comments on the Quality of English Language

Minor changes in the Discussion would improve the quality

Author Response

Interesting  study of a cohort of patients that had failed previous eradication treatment for H. pylori infection. Bacterial resistance represents a major challenge in infection eradication, and it justifies such a study.

We thank the referee for the kind appreciation of our study.

The remarkable low prevalence of resistance in this population as noted by the authors requires some additional comments. First, poor patient compliance could explain some of these findings. For instance, if treatment with one or more of the antibiotics use was stopped too early this could paradoxically explain the relative low resistance percentage to a given agent. Second, the resistance was detected, as specified in Methods by real-time PCR followed by high resolution melting (HRM) to detect mutations. Genotypic resistance is not synonymous of phenotypic resistance as it is not always expressed. Perhaps, authors could comment about this possibility in the Discussion.

We perfectly agree with the comments of the reviewer. Indeed, the quite low prevalence of secondary resistance may have many explanations. First, three out of 50 patients had a previous early therapy termination due to side effects (this additional result has been added); therefore, in these patients a real secondary resistance may not have occurred. Another reason could be the molecular biology method. PCR with HRM detects every variation in genetic sequence, but, differently from sequencing (NGS), it does not precisely define the point mutation; therefore, it is less accurate (Nezami BG, Jani M, Alouani D, Rhoads DD, Sadri N. Helicobacter pylori Mutations Detected by Next-Generation Sequencing in Formalin-Fixed, Paraffin-Embedded Gastric Biopsy Specimens Are Associated with Treatment Failure. J Clin Microbiol. 2019;57(7):e01834-18.  Egli K, Wagner K, Keller PM, Risch L, Risch M, Bodmer T. Comparison of the Diagnostic Performance of qPCR, Sanger Sequencing, and Whole-Genome Sequencing in Determining Clarithromycin and Levofloxacin Resistance in Helicobacter pylori. Front Cell Infect Microbiol. 2020;10:596371). However, NGS is more expensive and not available in all laboratories. On the basis of this concern, we preferred HRM, which may be rapid and cost effective for a large scale analysis. Finally, we perfectly agree that, among the limitations, it should be acknowledged that not all genetic mutations may translate into a phenotypic resistance, and we discussed also this point (Xiong M, Mohammed Aljaberi HS, Khalid Ansari N, Sun Y, Yin S, Nasifu L, Sun H, Xu T, Pan Y, Nie Z, Liu C, Zhang Z, Jiang Z, Wang S, He B. Phenotype and genotype analysis for Helicobacter pylori antibiotic resistance in outpatients: a retrospective study. Microbiol Spectr. 2023;11(5):e0055023).

Nothing is said of bismuth salts activity. H. pylori resistance to these agents is rare and colloidal bismuth subcitrate has been reported to prevent the development of H. pylori resistance to nitronidazole. It would be worth to highlight the importance of bismuth salts in a world of increasing antibiotic resistance.      

We agree with the comment of the referee. Therefore, in the discussion we added a paragraph stating that “bismuth salts are an effective weapon against H. pylori, as they have an intrinsic antibacterial effect, and they do not elicit any antimicrobial resistance. It has been demonstrated that adding bismuth to an eradication regimen may result in an additional 30%-40% gain to the success with resistant infections (Dore MP, Lu H, Graham DY. Role of bismuth in improving Helicobacter pylori eradication with triple therapy. Gut. 2016 May;65(5):870-8). Therefore, when available, adding bismuth should be always considered when facing difficult to eradicate H. pylori”.

Reviewer 2 Report

Comments and Suggestions for Authors

I can’t understand the results of the study: I counted that if 50 patients were enrolled and 72% failed one antibiotic course, 16% two, 10% three and 2% 4 courses that makes 71 failed courses. In the next sentence the authors write what were the numbers of particular eradication regiments failed but they enumerate only 35 cases. Are there the data about the rest?

Next the authors write about the number of cases of secondary antibiotic resistance. Again, it is difficult to understand the data. For example, in table 1 the number of amoxycillin resistant strands is 4 and but in the text it is 7 In line 79, 9 in line 80 and 8 in line 81. Maybe adding the column titled “total number of patients with antibiotic resistance” would help a little.  

Results presentation must be improved.

Low secondary antibiotic resistance in the present study is opposite to previous studies. Authors should discuss why. Is it caused by the DNA test used or rather different material? Is it possible that not all mutations causing resistance are diagnosed by this DNA test? Other cites studies tested the resistance in isolated strains and not in stool.

Comments on the Quality of English Language

English quality is quite good.

Author Response

I can’t understand the results of the study: I counted that if 50 patients were enrolled and 72% failed one antibiotic course, 16% two, 10% three and 2% 4 courses that makes 71 failed courses.

This result is explained by the fact a single patient may have failed more than one previous therapeutic regimen.

In the next sentence the authors write what were the numbers of particular eradication regiments failed but they enumerate only 35 cases. Are there the data about the rest?

In the text, we enumerated only the most commonly used regimens. Some other patients had undergone other therapies non listed by current guidelines (e.g. triple therapy with only amoxicillin and tetracycline or metronidazole, sequential therapy with levofloxacin), while some patients did not recall previous therapies.

Next the authors write about the number of cases of secondary antibiotic resistance. Again, it is difficult to understand the data. For example, in table 1 the number of amoxycillin resistant strands is 4 and but in the text it is 7 In line 79, 9 in line 80 and 8 in line 81.

Maybe adding the column titled “total number of patients with antibiotic resistance” would help a little. 

Results presentation must be improved.

The total number of amoxicillin resistant strain is 7. On the other side in the table we indicated the number of amoxicillin resistant strains among all patients who had a previous failure to a regimen containing amoxicillin.

Low secondary antibiotic resistance in the present study is opposite to previous studies. Authors should discuss why. Is it caused by the DNA test used or rather different material? Is it possible that not all mutations causing resistance are diagnosed by this DNA test? Other cites studies tested the resistance in isolated strains and not in stool.

We perfectly agree with the comments of the reviewer. Indeed, this point was also raised by reviewer 1. The quite low prevalence of secondary resistance may have many explanations. First, three out of 50 patients had a previous early therapy termination due to side effects (this additional result has been added); therefore, in these patients a real secondary resistance may not have occurred. Another reason could be the molecular biology method. PCR with HRM detects every variation in genetic sequence, but, differently from sequencing (NGS), it does not precisely define the point mutation, therefore it is less accurate (Nezami BG, Jani M, Alouani D, Rhoads DD, Sadri N. Helicobacter pylori Mutations Detected by Next-Generation Sequencing in Formalin-Fixed, Paraffin-Embedded Gastric Biopsy Specimens Are Associated with Treatment Failure. J Clin Microbiol. 2019;57(7):e01834-18.  Egli K, Wagner K, Keller PM, Risch L, Risch M, Bodmer T. Comparison of the Diagnostic Performance of qPCR, Sanger Sequencing, and Whole-Genome Sequencing in Determining Clarithromycin and Levofloxacin Resistance in Helicobacter pylori. Front Cell Infect Microbiol. 2020;10:596371). However, NGS is more expansive and not available in all laboratories. On the basis of this concern, we preferred HRM, which may be rapid and cost effective for a large scale analysis.